# communications
## engineering

# *Agave sisalana*: towards distributed manufacturing of absorbent media for menstrual pads in semi-arid regions

Anton Molina[1,2,4], Anesta Kothari [2,4], Alex Odundo[3] & Manu Prakash [2✉]

Agaves are robust, drought tolerant plants that have been cultivated for their high-strength fibers for centuries and they hold promise as a crop in the face of increasing water scarcity associated with a warming planet. Meanwhile, millions of women lack access to sanitary products to safely manage their menstruation particularly in low- and middle-income countries characterized by a dry climate. To address this issue, we show a processing route that transforms the leaves of the succulent *Agave sisalana* into a highly absorbent and retentive (23 g/g) material. The process involves delignification combined with mechanical fluffing to increase affinity for water and porosity, respectively. This process leads to a material with an absorption capacity exceeding those found in commercially available products such as menstrual pads. Finally, the carbon footprint and water usage associated with this process are comparable to those with common alternatives with the added benefit that it can be carried out at small scales while remaining environmentally sustainable. Our work represents a step towards distributed manufacturing of essential health and hygiene products based on a local bioeconomy.

[1] Department of Materials Science and Engineering, Stanford University, 496 Lomita Mall, Stanford, CA 94305, USA. [2] Department of Bioengineering, Stanford University, 443 Via Ortega, Stanford, CA 94305, USA. [3] Olex Techno Enterprises, Kisumu, Kenya. [4]These authors contributed equally: Anton Molina, Anesta Kothari. ✉email: manup@stanford.edu

Absorption materials are critical for a variety of items essential to basic quality of life such as bandages, diapers, and menstrual hygiene products. There is a large gap in the availability of menstrual hygiene products across the world. It is estimated that nearly 500 million women lack access to menstrual hygiene products[1]. In the absence of appropriate solutions such as disposable sanitary napkins, menstruating women often resort to improvised solutions which may pose a health risk or are forced into non-participation which results in unequal economic outcomes[2]. These negative outcomes associated with inadequate means to manage menstruation are often referred to as period poverty[3]. The use of improvised solutions such as cloth rags are particularly prevalent in rural settings (Fig. 1a)[4], where imported products are unable to achieve last mile distribution or are otherwise prohibitively expensive[1] (Supplementary Note 1, Supplementary Fig. 1).

Distributed manufacturing is a mode of production that occurs in close geographic proximity to the communities being served[5]. Reduced economies of scale are compensated by more resilient supply chains, simplified logistics, increased sustainability, and a capacity for network effects enabled by global sharing of know-how using digital communication technologies[6–8]. Distributed production of disposable menstrual pads is emerging as a promising route for addressing several shortcomings associated with relying on imported products, especially in serving the needs of rural communities[9]. These entities are often challenged by access to quality raw materials[10,11]. This challenge can be compounded by local geographic circumstances, for example water intensive production in semi-arid climates[12,13]. Recent advances in absorbent fiber materials have focused on materials produced from cellulose derivatives[14,15]. But these approaches require access to advanced timber products which might not be reliably available. Meanwhile, superabsorbent polymers (SAPs) are often included in absorbent hygiene products to increase absorption and retention while reducing the product's bulk[16,17]. SAPs produced from synthetic polymers are burdened by a high environmental footprint[18,19]. Advances have been made in the production of biodegradable SAPs using proteins sourced from agricultural waste streams[20,21]. Even so, to make a complete menstrual pad, these biodegradable SAPs must still be embedded in a fibrous matrix to provide structure and facilitate liquid transport[17,22]. Taken together with the other materials required for producing a disposable menstrual pad (e.g. porous top layer and waterproof back layer), conventional products represent a considerable sustainability challenge in terms of plastic waste[19,23], health effects[24], and their burden on sanitation systems[1,25]. Thus a key challenge in realizing the distributed production of menstrual pads will be to develop the necessary functional materials for their construction utilizing locally-sourced regenerative materials[26].

Typically, the fibrous matrix and key functional material in the vast majority of disposable menstrual pads is a fluff pulp composed of cellulosic fibers derived from wood. In general, wood resources are regarded as a promising, renewable replacement for many petroleum-based products[27]. Less discussed is that wood resources are distributed unequally across the planet (Fig. 1b)[28] and threatened by climate change[29,30]. Furthermore, wood fibers are extracted from chips using a harsh chemical treatment known as the Kraft process[31]. This process can only be implemented in an economically and environmentally sustainable manner with high material throughput (400-1500 kton/year)[32]. This combination of unequally distributed resources manufactured in a large-scale, centralized manner introduces supply chain fragility and leads to unequal access to downstream products[33,34]. Non-wood feedstocks have long been considered as alternative feedstocks for the pulp and paper industry. However, challenges associated with consistent feedstock supply, higher cost of

transport due to low density, and the need to adjust the Kraft chemistry due to different chemical compositions of the feedstocks have hindered adoption at large scales[35,36]. In support of these efforts, comparative studies of non-wood biomass have focused on properties predictive of performance in paper making[37,38], leaving properties relevant for niche applications such as absorption largely unexplored[39,40]. Nonetheless, small-scale pulping of non-wood alternatives have found application as absorbent materials for use in the production of disposable menstrual pads by small- and medium-sized entities[11]. While many of these efforts have struggled to scale and suffer from inconsistent or poor quality, they have demonstrated that meaningful social and environmental impact can be made by these emerging micro-pulping facilities ( ~ 1 ton/year)[1,11,24,41].

Biology provides us with examples of efficient delignification operating at the organismic scale. Wood-eating termites and wood-rot fungi represent "powerful mills that reduce ligneous food to a pulpy condition"[42]. While the mechanisms by which these systems operate are not fully understood, biological delignification is a multi-enzymatic process mediated[43] by diffusible small molecules[44]. The separation of cellulose from the lignin-rich binding matrix does not require complete degradation of the lignin. Non-enzymatic processes such as Fenton chemistry might be sufficient to disrupt crosslinks between cellulosic fibers and this binding matrix[45]. Recently, Fenton chemistry has been applied to biorefining[46], wastewater treatment[47,48], and as a chlorine-free alternative to bleaching in the pulp and paper industry[49]. In particular, the decomposition of organic peracids into reactive carbon-centered radicals enables an increased reactivity towards organic material compared to inorganic peroxides[50,51]. In the case of peroxyformic acid, decomposition occurs rapidly into water and carbon dioxide, eliminating the introduction of adsorbable organic halides into the environment[52]. Meanwhile synthetic systems that allow for recycling of reagents have also emerged based on solid dicarboxylic acids[53], deep eutectic solvents[54], and organosolv pulping[38,55]. Thus, there are two conceptual approaches to operating the pulping process that minimize dependence on an external chemical supply chain: recycling and on-site production. If the reagents can be recycled, energy must be expended to recover them. If reagents are consumed, then they must be efficiently produced on-site. Our study is motivated by the increasing capacity for on-site production of chemicals like hydrogen peroxide[48,56] and formic acid[57–59]. Few studies have investigated the use of these technologies to implement a bioinspired strategy to transform lignocellulosic biomass into absorbent media in an environmentally sustainable way at small scales.

To date, most local pad manufacturing efforts are built around the use of a limited number of plants, with banana pseudostems being one of the most common[60,61]. However, a reliance on a small number of plants reduces the ability of this type of manufacturing to extend into different geographies with distinct biomes. With this in mind, Agave sisalana (Sisal) is a promising candidate (Fig. 1c). Sisal is a robust and drought tolerant plant[62] traditionally used in the manufacture of cordage due to its strength and durability. Despite these attractive features, global production has been in decline since the introduction of synthetic fibers[63,64]. However, it has received renewed attention for its hardiness and potential to serve as a commodity crop in dry climates or on otherwise marginal lands. The biological basis for its success is rooted in the plant's crassulacean metabolism (CAM)[65]. While nearly 7 % of all plants use the CAM mechanism, the majority of them are small and lack any obvious economic value. Even though sisal has been used as a fiber feedstock in semi-arid regions for centuries, current applications remain limited. Expanding our capacity for obtaining useful materials

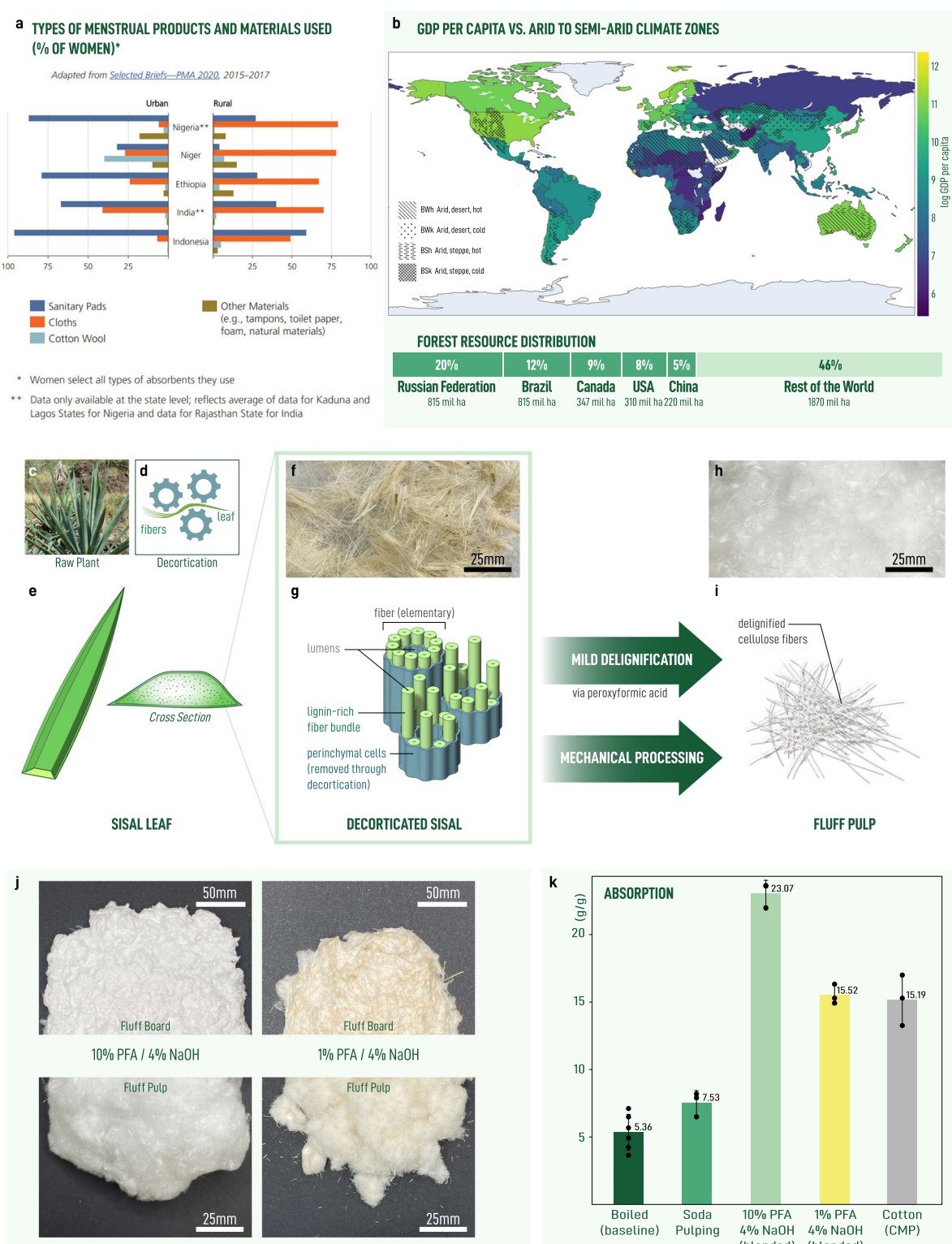

**Fig. 1 Sisal as a non-wood alternative to produce absorbent material in semi-arid regions to address period poverty. a** Preferred menstrual products in select low- and middle-income countries comparing urban and rural populations with data collected from ref. [1] (CC BY-ND 4.0). **b** Low per capita GDP is coincident with arid to semi-arid climate zones. See Supplementary Notes 1 for details on (**a**, **b**). **c** Photograph of mature sisal plant was sourced from "Sisal (Agave sisalana)" by Forest & Kim Starr CC BY 3.0. **d** Schematic of mechanical decortication device. **e** Illustration showing sisal leaf and cross section. **f** Photograph of dried macrofibers obtained from leaf via decortication. **g** Schematic of decorticated macrofibers in **f** showing its hierarchical strcuture. **h** Photograph showing fluff pulp obtained from decorticated macrofibers via mild delignification with peroxyformic acid and mechanical treatment. **i** Illustration of fibers shown in **h**. **j** Large-scale production of fluff pulp after treatment with 10% (left) and 1% (right) peroxyformic acid. **k** Absorption performance of sisal-based materials compared with cotton found in commercial menstrual products (cotton-CMP). Error bars correspond to one standard deviation.

from draught resistant commodity crops is critical for adapting to a warming planet and associated changes in biomass distribution. To our knowledge, an efficient demonstration of fiber extraction and use in absorbent applications has not been demonstrated.

Here, we apply a mild delignification chemistry based on peroxyformic acid to obtain absorbent microfibers from sisal that exceed the performance of cotton from commercially-available menstrual pads (cotton-CMP). We characterize the physical and chemical properties to understand structure-function relationships across multiple length scales. Finally, we perform a carbon and water footprint analysis of the manufacturing process and compare it with common alternatives showing that this strategy represents a route towards reductions in greenhouse gas emissions. These results represent a sustainable solution for providing access to high-quality absorption materials in semi-arid regions to enable local downstream production of sanitary napkins and other personal hygiene products. By developing a strategy for responsible manufacture of products aimed at reducing gender inequality, our work addresses several of the 17 Sustainable Development Goals described by the United Nations.

## Results and discussion

**Design criteria and evaluation**. A few criteria must be satisfied in order to obtain an absorbent and retentive material from a disordered network of fibers. Specifically, absorption in a fiber network depends on two factors: fiber-liquid interactions and the geometric structure of the fiber network[66]. The maximum absorption capacity of a porous media is a function of its porosity $\phi$:

$$A = \frac{\rho_l}{\rho} \frac{\phi}{1-\phi} \qquad (1)$$

where $\rho_l$ is the density of the absorbed liquid and $\rho$ is the density of the material from which the absorbent media is composed[67]. However, porosity alone is not sufficient. The material must also imbibe and retain liquid. The Lucas-Washburn equation can describe the ability of porous media to imbibe liquid by considering capillary flow in a bundle of cylindrical tubes. The penetration length $L$ of a liquid with surface tension $\gamma$ and dynamic viscosity $\eta$ is a function of time given by:

$$L(t) = \sqrt{\frac{\gamma r cos(\theta)}{2\eta}} t^{1/2} \qquad (2)$$

where $\theta$ is the contact angle between the imbibed liquid and the solid and $r$ is the pore radius. This analysis shows that the desired material will require a low contact angle. The role of pore size is more complicated, since larger pores will allow for fast mass flux as indicated by the Washburn equation while smaller pores will allow for a greater capillary pressure and will likely be associated with a higher surface area. A greater hydrophilic surface area will help the material retain the imbibed fluid when subject to an external load, a common scenario for most applications. Thus, the material must have a low contact angle, high porosity, and a high fraction of small pores.

To accommodate this complexity in geometry and chemistry, we measure the retention of an absorbed viscous test liquid under an applied load as a means for evaluating materials produced from biomass following the Scandinavian standard SCAN-C-33-80 (Supplementary Video 1)[39]. Fibers are compressed into a template with area 10.08 $cm^2$. Specimen height is often sensitive to how the specimen is dried. The viscous test solution is composed of a mixture of glycerol and water. This solution is designed to mimic the rheological properties of blood and was prepared according to IS 5405:1980 (Supplementary Note 2, Supplementary Fig. 2)[68]. We note that there is a gap in material

testing regarding the complex rheological properties of blood[69] that has only recently been addressed[70]. In what follows, we characterizes the physical and chemical transformations that underlie the conversion of sisal leaves into a highly absorbent material.

**Delignification of sisal fibers**. A challenge in working with non-wood feedstocks is that the properties of the fiber depend not only on the growing conditions but also on how the fibers have been harvested and subsequently extracted from the surrounding plant tissue primarily composed of parenchymal cells. For example, depending on the nature of the tissue, fibers can be extracted mechanically[71] through decortication (Fig. 1d, Supplementary Fig. 3) or enzymatically by retting[72]. Furthermore, care must be taken during decortication[73] to ensure that structural defects (e.g. kinks) do not accumulate on the fibers which might affect downstream absorption properties[40]. The fibers in this work are obtained by decortication. The waxy, lance shaped leaf (Fig. 1e) is fed by hand into a decorticator (Supplementary Video 2) and subsequently dried to yield our feedstock fibers (Fig. 1f). These feedstock fibers are hierarchically-structured, lignin-rich macrofibers fibers held together by an amorphous binding matrix composed of lignin, hemicellulose, and proteins such as pectin (Fig. 1g). In order to be useful in absorbent applications, a substantial fraction of the hydrophobic lignin must separated from the hydrophilic, cellulosic fibers in order to ensure uptake and retention of liquid. We begin by demonstrating that absorbent media cannot be obtained from sisal subject to mild soda pulping conditions (Supplementary Fig. 4). The resulting fibers show only a slight increase in absorption capacity (7.53 g/g) compared with the untreated reference (5.35 g/g) (Supplementary Note 3). Further, the fibers retain an uncomfortable coarse texture which make them unsuitable for application in a sanitary pad.

In order to efficiently separate the cellulosic fibers from the lignin-rich binding matrix, we first treat the fibers with in situ prepared peroxyformic acid (50 °C) followed by an alkali wash (50 °C) (Supplementary Video 3)[74,75]. Treatment with a peroxyformic acid solution can be regarded as a mild delignification which selectively removes lignin while preserving the structure of the cellulose microfibers. As noted in earlier work, this process is considerably less energy intensive than conventional delignification procedures which occur at high temperatures (80–160 °C) and pressures (for example, ≥0.5 MPa). The acidic conditions provided by the presence of aqueous formic acid will allow for hydrolysis of ether linkages leading to a reduced molecular weight of lignin and increased solubility[76,77]. Additionally, under acidic conditions, peroxides and peracids are powerful oxidizing agents that react electrophilically with electron-rich aromatic and olefinic structures[78] to produce carboxylic acids.

Following the oxidation with an organic peracid under acidic conditions, an extended sodium hydroxide wash acts to solubilize the cleaved and carboxylated lignin fragments. Meanwhile, hemicelluloses are generally regarded as soluble under alkali conditions where hydroxyl radicals can interrupt the hydrogen bonding of these branched polymers[79]. The exact details of the reaction pathways are difficult to know and are out of the scope of the present research, since they depend on both the identity of the reactive oxygen species and the composition of the lignin involved[50,78]. High concentrations (10%) of peroxyformic acid yield sisal microfibers that appear completely bleached with a yield of 60% (w/w) (Fig. 1h, i, and Supplementary Fig. 5) and high absorption performance (23.07 $g/g$). Lower concentrations (1%) of peroxyformic acid gave microfibers that retain some brown color with a yield of 77% (w/w) (Supplementary Fig. 6). The color and higher yields suggest appreciable amounts of

residual lignin are retained by the material (Fig. 1j). Despite the presence of residual lignin, the absorption performance (15.52 $g/g$), while reduced compared with the totally delignified samples, is still competitive with cotton-CMP (15.19 $g/g$) and bleached wood-derived fluff pulps (15.69 $g/g$) (Fig. 1k and Supplementary Figure 7).

To test our understanding of what is happening during each step (Fig. 2a), we characterize the structure of the resulting fibers using scanning electron microscopy (SEM) (Fig. 2b-d). We see that unprocessed sisal fibers are large with diameters ~ 100–250 µm (Fig. 2b). These large fibers are characterized by a rough surface in part due to debris from parenchymal cells[80]. Following treatment with the peroxyformic acid solution, we observe the macrofibers beginning to debundle into smaller constituents (Fig. 2c). Debundling occurs along both the radial and axial dimensions of the fibers. This suggests that the polymers composing the binding matrix have undergone cleavage. However, this process is only partial and we also observe a surface roughness which we attributed to residue from the material (lignin, hemicellulose, and proteins) continuing to bind fibers into these larger bundles. Finally, following the sodium hydroxide wash step and after mechanical agitation during washing, we observe nearly complete debundling of the macrofibers into their constituent microfibers (Fig. 2d). Close inspection of the microfibers shows that they have a smooth surface, suggesting that the lignin composing the binding matrix has been solubilized under alkali conditions. These microscopic changes are clearly seen at the macroscale where fiber specimens become increasingly white and composed of finer fibers (Fig. 2e–g).

We characterize the underlying chemical changes using Fourier transform infrared (FTIR) spectroscopy (Fig. 2h). FTIR spectra show the elimination of the peaks at 1240 and 1510 cm$^{-1}$ following treatment with peroxyformic acid. These peaks correspond to C-O, C-C stretch and aromatic ring vibrations in lignin, respectively[81]. The removal of these peaks is evidence for hydrolysis of ether linkages and oxidative ring openings in lignin. In particular, reactive oxygen species during treatment with peroxyformic acid generate carboxylic acids as a result of oxidative ring opening which then become highly soluble in alkaline solution. This interpretation is supported by the elimination of the peak at 1730 cm$^{-1}$ corresponding to C=O carbonyl stretching mode[82]. Cotton is used as a reference since its composition is nearly pure cellulose[83]. Comparison with cotton-CMP shows that our resulting material closely resembles cellulose. Taken together, these results are consistent with the picture established from SEM characterization.

In order to gain insight into how these different treatment steps affect the wetting properties of the fibers, we performed static angle tensiometry using the Wilhelmy principle which relates wettability $W$ to contact angle according to $W = cos(\theta)/\gamma$, where $\theta$ is the contact angle and $\gamma$ is the surface tension of water and wettability is the force of the submerged fiber ($F$) normalized by the fiber perimeter ($P$) (Fig. 2i), (Supplementary Fig. 8)[84]. Measurement of boiled sisal showed moderately hydrophilic material with contact angle of 46° whereas treatment with peroxyformic acid and sodium hydroxide gave contact angles 32° and 24°, respectively. The contact angle of alkali treated fibers are equivalent to the contact angle obtained from cotton-CMP, providing confirmation of the FTIR analysis that the chemical properties of the sisal microfibers is nearly pure cellulose. While contact angle measurements of natural fibers do have severe limitations as described in the literature[85], these results indicate a progressive increase in the hydrophilicity of the fiber surface with each step.

With a reduction in contact angle, we would expect an increasing trend in absorption for samples prepared from the

treated materials (Fig. 2j). This idea was tested by preparing samples from material after each treatment step and letting them air dry. Interestingly, there is not a significant increase in absorption following treatment with peroxyformic acid alone. However, there is an appreciable increase following alkali treatment, which is evidenced by the increasing intensity of absorbed test liquid following immersion such that the alkali-washed sample appears nearly black (Fig. 2e–g). At this point, there is still a significant gap between the alkali treated samples and the performance of cotton-CMP despite very similar physical and chemical properties of individual fibers.

**Structure of fluff pulp materials**. The capacity of a fibrous material to absorb and retain liquid depends not just on the properties of individual fibers but also on their mesoscale structure[66]. Porosity and pore size distribution are generally regarded as key structural factors at the network scale which determine absorption performance[67]. The relationship described in Eq. (1) suggests that to increase absorption capacity, increasing porosity will be key. However, the challenge in producing porous materials from wet fiber building blocks is network collapse and fiber-fiber bonding due to capillary forces generated during evaporation, causing a reduction in porosity and non-trivial changes in the pore-size distribution (Fig. 3a). The goal is to eliminate irreversible binding that would reduce the absorption and retention of the material (3b).

There are two strategies to obtain highly porous materials from wet fibers. The first is to eliminate capillarity during drying. This can be realized by freeze drying, that is to replace evaporation with sublimation. The second is to disrupt any structures that do form through a mechanical process like blending, milling, or pulverizing. The later is practiced industrially using a hammer-mill where chemical debonding and anti-static agents are often added to reduce energy costs and increase yields. Here, we compare the performance of samples prepared by freeze drying with blending implemented using a benchtop blender without the use of any chemical agents. We can see that blending substantially increases the absorption capacity of the materials resulting in an increase from 9.64 $g/g$ to 23.94 $g/g$ (Fig. 3c). To verify that porosity is a key control parameter in this process, we plot absorption for all of the samples prepared as a function of porosity (Fig. 3d). In general, good agreement with Eq. (1) speaks to the high capacity for liquid retention since the curve represents the theoretical maximum for a liquid with $\rho_l = 1.08 g/mL$.

To better understand the effects of these bulk processing steps, we compare the structural features of samples before (air dry only) and after dry blending. At the macroscale, air dried samples are stiff, brittle, and show the formation of layers (Fig. 3e). After blending, the layering and brittleness are eliminated and we obtain a material that is soft to the touch (Fig. 3f). An additional comparison with a sample prepared from 1% peroxyformic acid-treated fibers shows that blending of these fibers creates a material similar to that obtained from 10% fibers (Fig. 3g). Inspection of SEM micrographs shows that the layered structures in air dried samples are effectively broken up after blending (Fig. 3h–j). We complement these observations with X-ray computer microtomography ($\mu$CT) images of samples before (Fig. 3k) and after dry blending (Fig. 3l, m) at the millimeter scale. Analysis of the unblended samples shows a small number of macrofibers with diameters ~ 100 µm dispersed in a random matrix of microfibers (~ 20 µm). We suspect some fraction of these fiber bundles exist after the peroxyformic acid treatment; however, their extent might also be increased by capillary adhesion. In contrast, these bundles are entirely eliminated following dry blending. The dry blended samples consists of an isotropic distribution of single

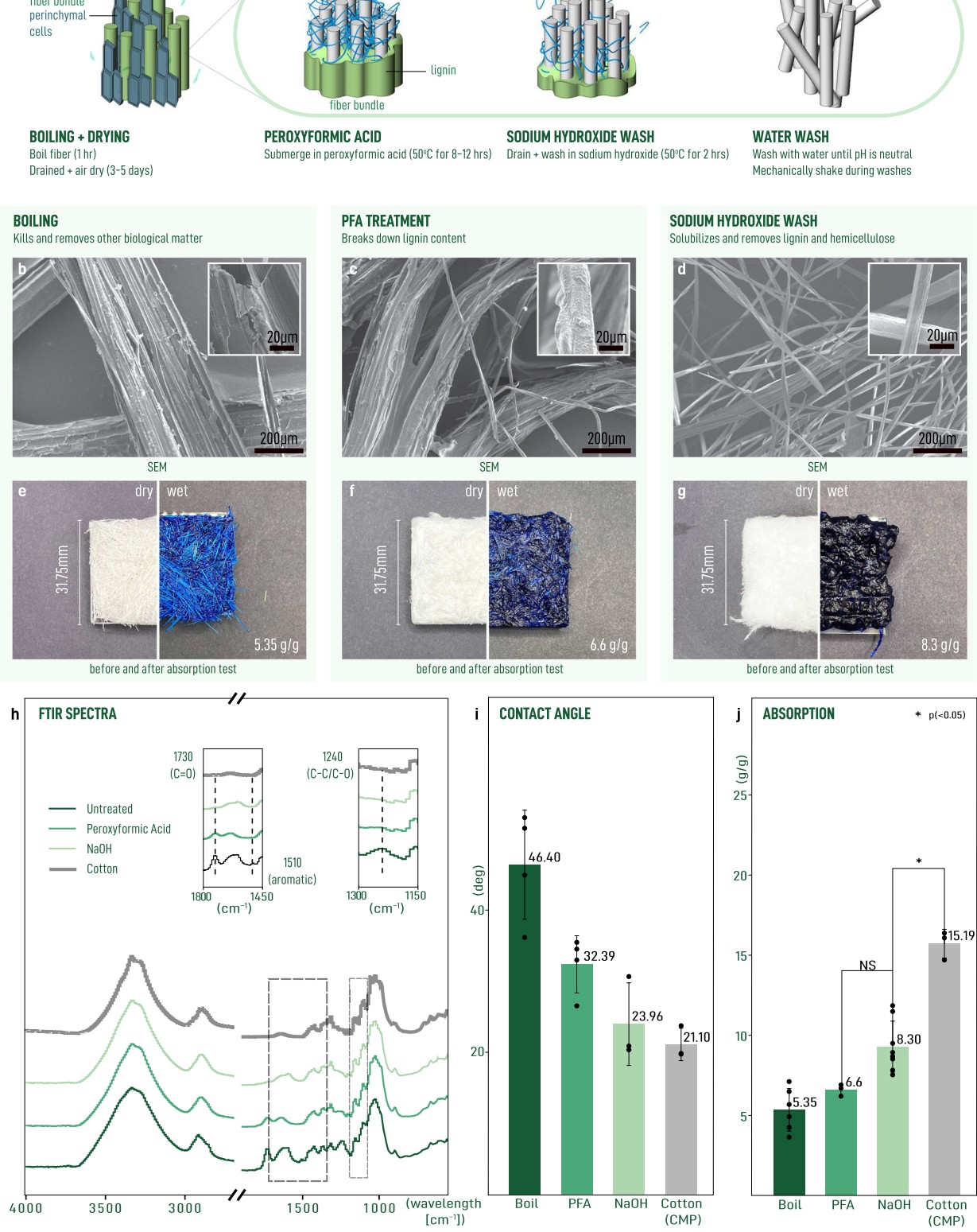

**Fig. 2 Evolution of structural and wetting properties from decorticated macrofibrers to delignified microfibers. a** Schematic of the delignification process showing reaction of peroxyformic acid with sisal macrofibers, subsequent solubilization under alkali conditions, and debundling during aqueous wash step. SEM micrographs of fibers after boiling **b**, treatment with peroxyformic acid **c**, wash with sodium hydroxide and water **d**. **e–g** Photographs showing test squares prepared from **b–d** before (left) and after (right) absorption testing. The darkness of the color correlates to the amount of fluid retained by the sample. **h** FTIR spectra with important peak changes annotated in insets. **i** Advancing contact angles of individual fibers obtained from static contact angle tensiometry. **j** Absorption performance for test squares prepared from fibers. Student's *t* test *P* value > 0.05, NS (not significant); *$P$ value < 0.05. Error bars correspond to one standard deviation.

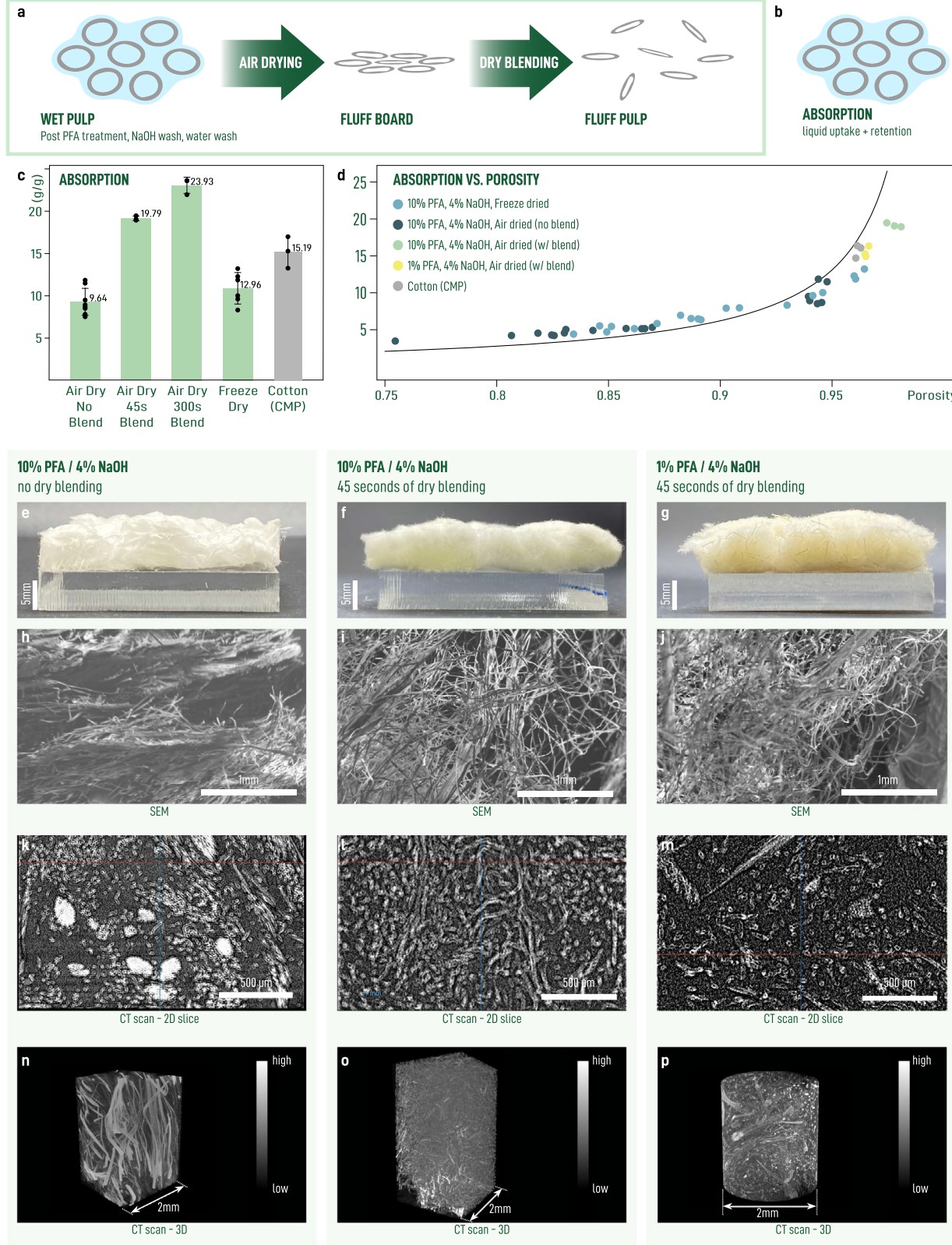

**Fig. 3 Increasing absorption capacity through mechanical processing. a** Schematic showing collapse of network during drying and subsequent debonding of collapsed network into porous fluff pulp. **b** Schematic showing how debonding correlates with liquid uptake and retention. The extent of debonding depends on the duration of dry blending. **c** Absorption capacity for test squares prepared from samples subject to different processes of drying and mechanical treatment. Error bars correspond to one standard deviation. **d** Absorption capacity for test squares with the black curve showing the theoretical absorption capacity as a function of porosity. Photographs showing cross-section of test squares for 10% perfooxyfomic-acid treated, air dried, unblended **e**, 10% perfooxyfomic-acid treated, air dried, blended (45 s) **f**, 1% perfooxyfomic-acid treated, air dried, blended **g**. SEM micrographs **h–j**, μCT cross sections **k–m**, and reconstructed 3D volumes **n–p** of samples shown in **e–g**, respectively.

microfibers. We note that the sample prepared from 1% peroxyformic acid-treated fibers retains a small amount of macrofibers, a reflection of less extensive delignification. Inspection of the reconstructed 3D volumes shows that this is accompanied by narrowing of the pore size distribution with the average shifted towards smaller pores (Fig. 3n–p), satisfying the criteria for a highly absorbent material.

**Comparison with other potential lignocellulosic feedstocks**. To see whether or not this process can be readily extended to other potentially abundant sources of lignocellulosic biomass, we apply this procedure to flax and hemp fibers (Fig. 4). At the macroscale, we observe that all samples have been converted to a white fibrous material. There is a clear difference in porosity between the two materials with flax-derived fluff pulp being much denser than the hemp-derived fluff pulp and both, in turn, being less dense than cotton-CMP (Fig. 4a–c). Comparison of structural features from SEM images of individual fibers before (Fig. 4d, e) and after (Fig. 4f, g) processing show similar debundling through delignification occurring for both fiber types. Both display an increased rough surface attributed to a second process of debundling, exposing 1 μm fibers. These features could pin the advancing contact line and reduce the uptake of fluid into the porous media. Further, we observe that flax fibers consist primarily of short, kinked fibers. We suspect that shorter fibers can pack more densely, resulting in a less porous and therefore less absorbent material. These features stand in contrast to the long smooth fibers obtained from cotton-CMP (Fig. 4h, i). FTIR characterization shows that the processing converts lignified material with distinct spectra into a material with similar functional group composition as cotton-CMP (Fig. 4j). Despite these functional group similarities, measurement of absorption performance follows the trend in density (Fig. 4k, l). While both flax and hemp underperform cotton-CMP and sisal, their performance does not preclude their use in absorbent applications. Taken together, these observations suggest that structural features at the single fiber level can play a decisive role in determining the performance of the resulting material.

**Life cycle carbon footprint analysis**. The unique biological characteristics of sisal make it and other members of the agave family promising candidates for building a bio-economy. Sisal can be harvested year round across a variety of geographies (Fig. 5a) yielding of over 200,000 tonnes globally in 2020[86]. Sisal is planted at densities of up to 3000–5000 plants per hectare and require 40–48 months before its leaves can first be harvested, after which a total of 50-60 leaves may be collected per year[87]. A typical leaf will contain 4% fibers by weight[88] and the plant will yield leaves until it is 9–12 years old[87,88]. *Agave sisalana* can yield 1.5 ton/hectare (whole leaf) whereas *Agave* hybrid 11648 can be harvested earlier and yield up to 2–3 ton/hectare (whole leaf)[64]. Meanwhile, other members of the agave family have shown even higher yields[62]. These figures show that sisal can be cultivated in large enough quantities to support the manufacture of consumer products in semi-arid regions. As a tangible demonstration, we incorporate sisal cellulose fluff pulp into a menstrual pad, borrowing top and bottom layers from a CMP (Fig. 5b–d).

To quantify the sustainability of this approach, we perform a cradle-to-gate carbon footprint life cycle analysis (LCA) to determine the impact of sisal cellulose microfiber production (Fig. 5e–g, Supplementary Note 4). The functional unit is 1 kg of sisal cellulose fluff pulp and the system boundaries include sisal cultivation, harvesting, manufacturing, and transportation. The life cycle inventory is shown in Supplementary Tables 1–2. Here, we consider two scenarios: (1) production of sisal as it occurs in

the lab and (2) an aspirational scenario where formic acid and hydrogen peroxide are produced on-site using commercially available electrocatalytic systems powered by renewable, solar energy[57,89]. Our analysis shows that there is a footprint of 1.195 and 3.475 kg $CO_2$-eq per kg sisal cellulose microfibers for the on-site and lab-scale production scenarios, respectively (Fig. 5f). This is comparable with reported values for the production of bleached cellulose fluff pulp derived from softwood timber resources (0.513–1.113 kg $CO_2$-eq) and from bleached cotton (1.65–5.25 kg $CO_2$-eq) (Supplementary Note 4).

Next, we show the carbon footprint associated with each production activity (Fig. 5g, Supplementary Note 5, Supplementary Tables 3–5). We see that the on-site scenario enables a reduction in the carbon footprint when powered by renewable energy. Further, this footprint compares favorably with competing alternative processes, especially cotton. The large footprint associated with cotton is associated with the amount of energy needed for upstream fertilizer production that is needed - if at all - to a far lesser by sisal or timber. This highlights the importance of considering the characteristics of the biomass feedstock used in the production of bio-based materials.

Since this work is motivated by small scale manufacturing efforts utilizing bio-mass based supply chains close to the point of use, we evaluate the transportation of materials after the gate of production (post-gate). More specifically, this means accounting for transportation of the fluff pulp from the fluff pulp processing facility to the pad manufacturing facility and transportation of the assembled product to the final market (Supplementary Note 6, Supplementary Tables 6–8). We neglect last mile distribution due to a paucity of high-quality information, despite it being well known as a major contributor of cost and greenhouse gas emissions[90]. For the different extended transportation scenarios considered, we find that transportation post-gate accounts for ~ 5–90% of the total carbon footprint. Again, these are conservative figures since they ignore last mile distribution. Thus, manufacturing built on localized supply chains has the potential to reduce this category of emissions.

In addition to carbon footprint analysis, we compare the direct water consumption for the process described in this work to water consumption of alternatives (Fig. 5h, Supplementary Note 7, Supplementary Table 9). The lab process is competitive in terms of total water consumption (44.6–119.6 kg $H_2O$ per kg fluff pulp) while the on-site production scenario requires additional $H_2O$ associated with on-site production of reagents (64.5–139.5 kg $H_2O$ per kg fluff pulp). The lower limit makes some conservatives assumptions about water reuse while the upper limit assumes no recycling. This is more than the water used in the production of softwood-derived fluff pulp which requires 61.78 kg $H_2O$ per kg fluff pulp[31,49], our process avoids the emission of adsorbable organic halides. Compared with water consumption data reported for a small-scale production facility located in Uganda, (90–300 kg $H_2O$ per kg absorbent)[11], our process represents an improvement. The amount of water consumed can be collected in semi-arid regions (250-500 *mm* rainfall/y), representing a required catchment area of 610–882 $m^2$ (Fig. 5i, Supplementary Note 8). This analysis supports the feasibility of fluff pulp production in semi-arid regions.

## Conclusions

This study made use of peroxyformic acid chemistry as a route towards extracting absorbent cellulosic microfibers from harvested sisal leaves. In general, peracid chemistry remains largely unexplored, even more so its application in delignification. The observation that partial delignification still results in good absorption and retention suggests that we can expect further

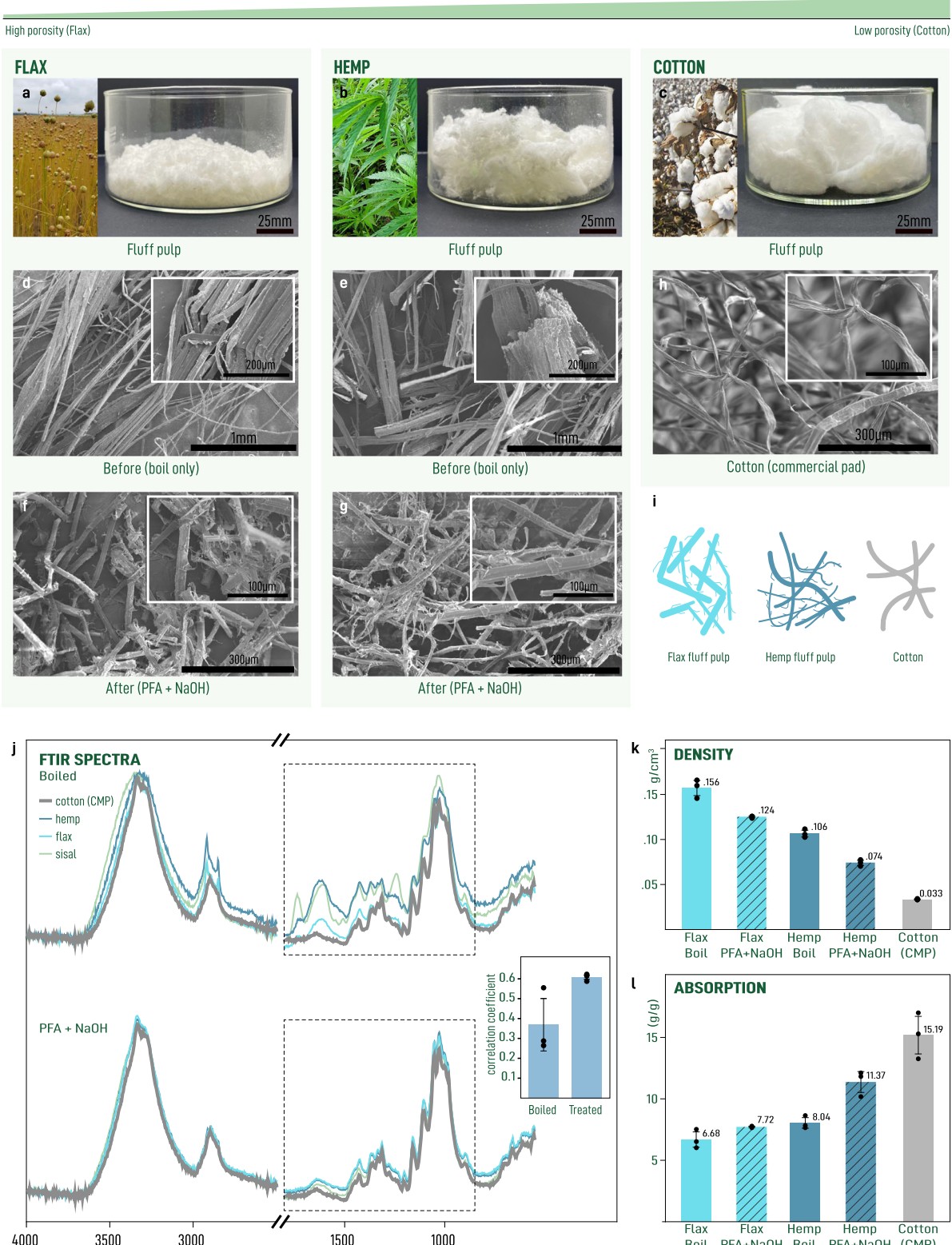

**Fig. 4 Morphology of single fibers from different fiber-rich plants effects absorption capacity.** Photographs of large-scale production of fluff pulps from flax **a**, hemp macrofibers **b**, and cotton collected from commercial menstrual products (cotton-CMP) **c**. SEM micrographs of boiled macrofibers **d**, **e** and fluff pulps (45 s dry blended) **f**, **g** obtained from flax and hemp, respectively. **h** SEM micrograph showing cotton-CMP. **i** Illustration highlighting the key morphological features of the fluff pulp and cotton-CMP fibers. **j** FTIR spectra comparing boiled macrofibers (top) with fluff pulp microfibers (bottom). Inset shows spectral correlation coefficient for each fiber type referenced against cotton-CMP. Densities **k** and absorption capacity **l** of test squares squares. All error bars correspond to one standard deviation. Reference photographs of flax **a**, hemp **b**, and cotton **c** plants were sourced from "Linen, Flax, Field image" by Tuikkis via Pixabay CC0 1.0 DEED, "Marijuana, Hemp, Cannabis image" by Mayya666 via Pixabay CC0 1.0 DEED, and "Cotton, Cotton field, White image" by jdblack via Pixabay CC0 1.0 DEED, respectively.

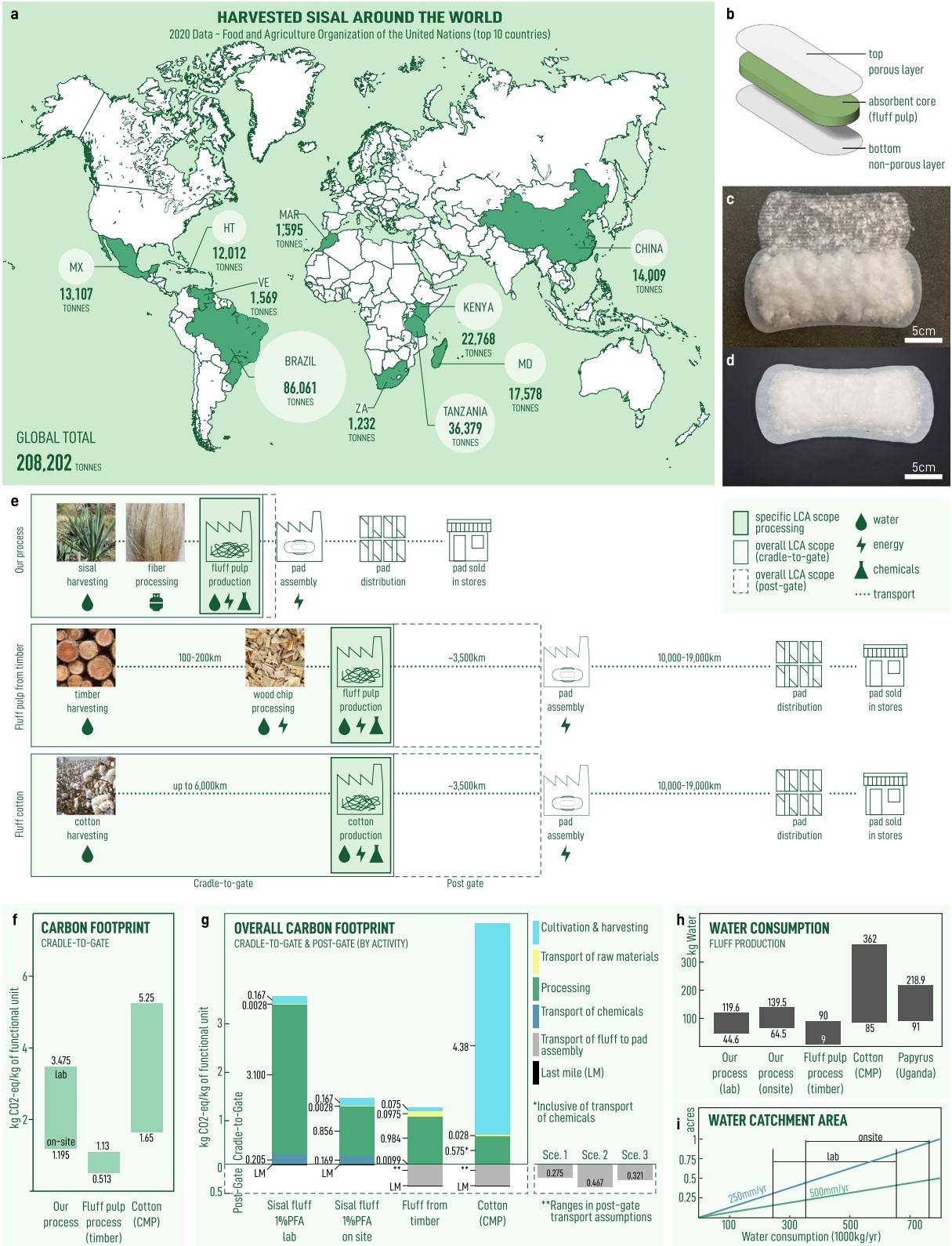

improvement in reagent consumption by controlling peracid concentrations and its activation. Meanwhile extraction of cellulose from lignocellulosic biomass generates a lignin-rich waste stream where the properties of the lignin are sensitive to the extraction chemistry. Developing strategies to valorize these waste streams will be important since they might enable the production of additional products. For example, waste lignin might be useful

for producing compostable barrier sheets[54] while other components might be useful as fertilizer[88]. The former will enable the production of complete products such as menstrual pads using regional biomass while the latter enables a regenerative bioeconomy[26]. More broadly, this is a chemical strategy that exists amongst many other possible alternatives such as recyclable chemicals, solid di-carboxylic acids, and enzymes. However,

**Fig. 5 Life-cycle analysis of sisal fluff pulp production in the context of distributed manufacturing of menstrual pads in semi-arid regions. a** Geographic distribution of the 10 largest nations producing sisal based on data obtained from the Food and Agriculture Organization (FAO)[86]. **b** Schematic showing assembly of a three-layer menstrual pad. Photographs showing sisal fluff pulp incorporated **c** into an assembled three-layer pad **d**. The other two layers are obtained from a commercially available menstrual pad. **e** Schematic showing activities described in life-cycle analysis with boxes defining the activities within each scope of the analysis. **f** Carbon footprint analysis of fluff production with comparison to common alternatives (wood-derived fluff pulp and cotton). Upper and lower bounds for our process correspond to lab and on-site production strategies, respectively; for common alternatives, these bounds are defined by values obtained from literature. **g** Carbon footprint analysis showing contribution by activity with cradle-to-gate and post-gate scopes. **\*\***Represents the three post-gate transportation scenarios considered (see Supplementary Note 6). **h** Water consumption associated with fluff production. **i** Water catchment area required to support production dependent on annual rainfall assumptions for arid and semi-arid regions (see Supplemental Note 8). Reference photographs of timber, wood chips, and cotton were sourced from "Timber stacked" by ID: 5903515 via Rawpixel CC0 1.0, "Wood chips background" by ID: 6015856 via Rawpixel CC0 1.0, and "Cotton, Cotton field, White image" by jdblack via Pixabay CC0 1.0 DEED respectively.

understanding these numerous trade offs must be considered within a comprehensive technoeconomic framework.

Despite similarities in functional group composition, the effectiveness of a given biomass feedstock varies depending on its structure. A better understanding of the relationship between biomass structure and processing is clearly needed. The complexity of these relationships highlights the importance of studying the connection between structure and performance in relatively niche applications such as absorption. A deeper understanding of these connections will enable the identification of lignocellulosic fiber feedstocks that are amenable to this processing strategy. When combined with an understanding of the geospatial distribution of biomass cultivation[91], this knowledge will enable similar production processes to be implemented in distinct climates and biomes.

Addressing the supply-side problem of period poverty requires innovation in materials and manufacturing methods[1]. Satisfying the growing demand for disposable menstrual pads with conventionally produced products represents a waste and sanitation burden. Existing manufacturing methods and associated supply chains have not succeeded in serving a large fraction of the planet's population, particularly those in rural low- and middle-income settings. Emerging small- and medium-scale manufacturers producing compostable menstrual pads from locally sourced biomass is an emerging production mode that addresses both concerns of environmental sustainability and product accessibility. In this work, we have shown that organic peracid chemistry can be employed for the partial delignification of sisal fibers to produce an absorbent material with a performance meeting and exceeding that of cotton-CMP. This demonstration will increase access to materials used in menstrual pads by paving the way toward their production in semi-arid regions. Furthermore, the carbon footprint of this process is competitive with conventional approaches particularly when manufacturing and product distribution exist within the same geographic scope. More broadly, with an increasing demand for timber resources driven by the transition towards the bio-economy, accessing lignocellulosic biomass from alternative sources particularly on dry or otherwise marginal lands will be increasingly important. Additionally, complementary production strategies are still urgently needed to realize the production of a complete menstrual pad from sustainable material feedstocks. Our work presents a manufacturing strategy for a key component of an essential product to be made in a distributed fashion, bringing further economic development in regions disadvantaged by climate.

## Materials and methods
**Materials**. Sisal leaves were obtained by Alex Odundo in Kisumu, Kenya. Hydrogen peroxide, formic acid, sodium hydroxide, sulfuric acid, iron sulfate, methyl paraben, methylene blue, potassium iodide, glycerol, and ethanol were obtained from Sigma

Aldrich and used without further modification. Gum arabic was obtained from Earthborn Elements.

**Decortication of sisal leaves**. In this work, sisal plants were obtained from the Nyanza region in Kenya. Sisal leaves (Fig. 1c, e) were harvested by hand before the fibers were extracted by decortication using a single-head decorticator (Olex Techno Enterprises) (Fig. 1d, Supplementary Video 2). The fibers were then cleaned and air dried (Fig. 1f and Supplementary Fig. 3). In the dried state, fibers are shelf stable for at least 1 year.

**Preparation of sisal cellulose microfibers**. Fibers were cut into short segments (~10 mm long) and boiled in water to remove any water soluble components. Following the procedure described previously, the stems were then delignified using 10% $v/v$ peroxyformic acid (synthesized in situ by combining 30% hydrogen peroxide with 95% formic acid in a 1:1 mole-to-mole ratio using 1% sulfuric acid as a catalyst) at 50 °C overnight[75]. The fibers were then treated with 4% ($w/v$) sodium hydroxide for 2 hours at 50 °C. In another realization, we followed the procedure describe by Haverty, et al.[46]: cut fibers were delignified using 1% ($v/v$) peroxyformic acid. In this instance, the amount of sulfuric acid was increased to 4%. The mixture was left to react for 45 mins before adding 400 μL of a solution containing $Fe_2(SO_4)_3$ (1 $kg/L$). This mixture was then left to react overnight at 50 °C overnight. The fibers were then treated with 4% ($w/v$) sodium hydroxide overnight at 50 °C. Finally, the fibers were washed with three times with equivalent volumes of deionized water or until the pH was neutral. It is important to note that the fibers were subject to approximately of 30 s of vigorous shaking during each wash (Supplementary Video 3). This introduction of mechanical energy is analogous to the use of Hollander beater used in conventional paper making and is important for defibrillating the fibers 100 $\mu m$ macrofiber bundles into smaller microfibers (in the form of wet pulp).

**Preparation of test squares**. For each test square, a standard acrylic template measuring 1.25"x1.25"x0.25", was used (Supplementary Fig. 2). Three methods were performed to produce various test square samples: (1) freeze drying, (2) air drying without blending, and (3) air drying with blending. For the freeze-dried samples, wet pulp was molded into the acrylic template and frozen at −80 °C overnight. The frozen samples were then transferred into a lyophilizer machine (7751000 Freeze Dry System, Labconco) and left to lyophilize over 2 days. For the air dried samples without blending, wet pulp was molded into the acrylic template and left to air dry for 3 days. For the air dried samples with blending, wet pulp was poured onto a mesh surface and left to dry for 3 days. The resulting fluff board was then cut into small pieces (~5cm x 5cm squares) and dry blended using a benchtop blender (700G Waring Blender). The resulting fluff

pulp was then molded into the acrylic template. Each sample was then removed from the template prior to the absorption testing.

**Absorption under pressure testing.** Absorption is assessed using a test solution prepared according to Indian Standard 5405:1980[68]. Briefly, 400 mg of methyl paraben and 74 g of gum arabica are added to 600 mL of boiling water. Once cooled, 900 mg of methylene blue and 147 mL of glycerin are added and the final volume is adjusted with water to 920 mL. The solution was mixed thoroughly and allowed to stand at least 24 h. Prior to the absorption test, each test square was weighed ($W_i$) and its average height measured using a caliper. The absorption under pressure (AUP) setup consists of a liquid bath, a porous transfer plate, a test square sample, a porous transfer plate, and a weight, as shown in (Supplementary Fig. 2). For each AUP, one test square was placed onto the first transfer plate, ensuring it makes contact with the liquid, before a second transfer plate and weight were placed on top (load of 50 $\frac{g}{cm^2}$). The test square is left to absorb for 5 minutes; longer absorption times were evaluated but absorption capacities after a longer absorption time did not excess 6% from the 5-minute absorption time. Then the stack of two transfer plates, test square, and the weight was lifted out of the solution and placed over a new container to drain excess liquid for 2.5 minutes. The wet test square is then weighed ($W_f$). We define absolute absorption in terms of an initial dry weight $W_i$ and final wet weight $W_f$:

$$absorption = \frac{W_f - W_i}{W_i} \tag{3}$$

All measurements are made in triplicate and the average value is reported.

**SEM.** A Hitachi S3400N SEM operated at 5 keV was used to obtain the micrographs. The compacted fiber samples were attached to the stage using conductive carbon tape and sputter coated with Au/ Pd (60:40 ratio).

**FTIR.** FTIR spectra were obtained using a FTIR spectrometer (Nicolet iS50 FT/IR Spectrometer) equipped with an attenuated total reflection (ATR) unit. Spectra were recorded with a resolution of 2.0 $cm^{-1}$ with 32 scans in the range of 4000 to 525 $cm^{-1}$. Quantitative comparison between different spectra were made using the second-derivative spectra correlation coefficient $r$ introduced by Presterleski et al.[92]:

$$r = \frac{\sum^N x_i y_i}{\sqrt{\sum x_i^2 \sum y_i^2}} \tag{4}$$

where $x_i$ and $y_i$ represent the intensity of second-derivative spectra at the $i$th frequency position. Spectral absorbance values in the fingerprint region between 800 and 1800 $cm^{-1}$ were used in this calculation. Normalization of spectra is not required for this calculation. Identical spectra will return a value $r = 1.0$.

**Static contact angle tensiometry.** All advancing wetting measurements were made following the method described previously by Young[84]. Briefly, individual fibers were pressed between a sheet of paper and fixed in place with a small amount of adhesive. The assembly is attached to a small wire hook coupled to a sensitive electrobalance (KSV Nima, Finland). The mounted fiber is lowered slightly above a petri dish filled with distilled water. A fixed amount of water is added to immerse a portion of the fiber. The force $F$ resulting from an increase in weight when the fiber is immersed in liquid is related to contact angle $\theta$ of the fiber by the Wilhelmy equation:

$$F = P\gamma_{LV}cos(\theta) \tag{5}$$

where $P$ is the perimeter of the fiber and $\gamma_{LV}$ is the surface tension at the liquid-vapor interface of water (72 $\mu N/m$), allowing for the determination of $\theta$. Prior to measurement, the diameter of each fiber is first recorded in the dry state with an optical microscope and used to determine the perimeter $P$ when calculating fiber wettability $W = F/P$. All measurements are made in triplicate and the average value is reported.

**Micro-computed X-ray tomography.** The 3D morphological analysis is conducted by micro-computed X-ray tomography ($\mu$CT) using an Xradia 520 Versa X-ray CT (Carl Zeiss, GmbH). Typical samples comprised 200 $\mu g$ material. Contrast of samples was increased by staining with 1% aqueous KI for at least 1 hour. In order to stain the blended samples while preserving their structure, these samples were freeze dried for 24 hours as described above. Samples are scanned with an accelerating voltage of 80 kV and 7 W power with no filter. 1600 projects over 180° angle of rotation. The geometric magnification $M_g$ is related to the source-to-object distance $d_{so}$ and object-to-scintillator distance $d_{os}$ by:

$$M_g = \frac{d_{so} + d_{os}}{d_{so}} \tag{6}$$

Large area scans were taken with a 0.4X objective and 1 s exposure with $d_{so} = 21.7\mu m$ and $d_{os} = 155.9\mu m$ fixed, giving $M_g = 8.18$. Small scan used to obtain information on individual fibers were taken with a 4X objective and 2.5 s exposure with $d_{so} = 21.7\mu m$ and $d_{os} = 50.0\mu m$ fixed, giving $M_g = 3.30$.

**Life cycle assessment.** A cradle-to-gate life cycle analysis was performed on sisal cellulose microfibers to quantify the materials carbon footprint. The LCA was performed according to ISO 14040[93] using SimaPro version 8.0.

## Data availability

The authors declare that the data supporting the findings of this study are available within the paper and its Supplementary Information files. Should any raw data files be needed, they are hosted with at Open Science Framework (https://doi.org/10.17605/OSF.IO/Z7M5A) and are otherwise available from the corresponding author upon request.

## Code availability

The code that support the plots within this paper and other findings of this study are available from the author upon reasonable request.

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

## Acknowledgements

We would like to thank Diego Brito for supplying the original banana pseudo stems used for preliminary experiments for this project. We would like to thank the team at Biolin Instrument for graciously performing preliminary dynamic contact angle measurements. We would like to thank Melanie Hannebelle and Ray Chang from the Prakash Lab for the time-lapse setup and the rheology measurement, respectively. We acknowledge all members of the Prakash Lab, the LGP2 group from the Université Grenoble Alpes, the NIDISI group, and the MitiMeth group for useful and exciting discussions. We would like to thank the SEM sample coating service provided by the Stanford Nano Shared Facilities. Part of this work was performed at the Stanford Nano Shared Facilities (SNSF), supported by the National Science Foundation under award ECCS-2026822. The work was financially supported by grant to M.P. from the Bill and Melinda Gates Foundation under award UAMBN (SPO-220079).

## Author contributions

A.M., A.K. are credited with investigation, methodology, formal analysis and writing the original manuscript. A.K. is credited with data visualization. A.O. provided key resources for the project and background context for the work. A.M, A.K, M.P. conceptualized the research, reviewed & edited the manuscript. M.P. acquired funding, supervised the research and validated the research.

## Competing interests

A.O. is founder of Olex TechnoEnterprises which designs and manufactures sisal decorticators. The remaining authors have no competing interests to declare.
