## [Peer Review File · Communications Engineering]

Reviewers' comments:

Reviewer #1 (Remarks to the Author):

The manuscript submitted by Molina et al. shows a promising novel source of absorbent alternatives based on Agave Sisalana structures. The work has a high potential to become a reference in sustainable absorbent materials using novel bio-based sources. However, some improvements are needed to increase the article's impact and clarify the work's aim. Please, find here my point-by-point comments:

- 1) In the title, what do the authors mean by "distributed production"? The title should be clear, reflecting the main outcome of the work.
- 2) In the abstract, the 23 g/g refers to absorption or retention? These two are very distinct terms in superabsorbents and need to be clear. Also, only absorption under load (AUL) was assessed, while other important parameters, such as free swelling and centrifuge retention capacity, are not stated. These evaluations are critical for proposing new materials to replace current absorbent items such as single-use pads. Please check the EDANA standards (nonwovens) for more information.
- 3) Is it unclear to the reviewer how the work will contribute to increasing access to menstruation pads for women in medium/low-income countries? This needs to be clearer in the revised version.
- 4) Non-wood alternatives are stated in the introduction. However, the AUL obtained in this study needs to be compared with other non-wood options and/or synthetic superabsorbent polymers (SAP) used commercially today. The authors should put the work into a comparative perspective to increase the impact further.
- 5) The authors mention non-wood alternatives in the introduction and discuss other lignocellulosic alternatives in the article text. However, what about other works pursuing alternative sources of absorbent polymers, e.g., proteins from agro-food biomass waste? A reflection on this could broaden the perspectives of the suggested materials. Some references are suggested here:
<https://onlinelibrary.wiley.com/doi/10.1002/jsfa.9738>
<https://doi.org/10.3390/polym15020351>
<https://doi.org/10.1038/s42004-021-00491-5>
- 6) How sustainable is the delignification process suggested? Are the reagents recirculated? How is the process thought on a large scale, and what are the possible side-streams generated? How cost-competitive are these materials compared to non-wood alternatives and synthetic SAP materials?
- 7) Line 272, again, the results are compared with bleached cellulose fluff pulp, but what about the absorption values of this material and also compared to synthetic counterparts?
- 8) Line 325, what is that sentence?
- 9) Line 383, which liquid are the authors referring to? The results can vary greatly between different

liquids of interest for the sanitary area. For instance, pure water vs. saline solution (usual industrial standard) can produce very different absorptions due to osmotic pressure effects. More information can be found at: <https://doi.org/10.1021/acssuschemeng.8b05400>

10) Line 384, is 5 min a representative value for absorption in sanitary applications? Normal values are around 30 min.

11) What is the intention behind assessing the static contact angle for an absorbing material?

12) Figure 2, why does the color of some materials after absorption look blue, black, and mixture?

13) Figure 2, FTIR is missing the X-axis units.

14) The scale of the microCT in Figure 3 needs to be included. Please add the scale bar.

Reviewer #2 (Remarks to the Author):

This manuscript evaluated the absorbing performance of delignified sisal fibers in the application of menstrual pads. The reviewer believes this study is very essential for the woman hygiene improvement and sustainability development in local communities. Minor revision is recommended prior to publication. Please see the comment details below.

(1) In Page 6 Line 144, it may be easier to understand when a detailed data is added here.

(2) In Page 7 Lines 159, please add citation here.

(3) In Page 7, Lines 163, maybe it will be better to add yield data here to help describe.

(4) In Page 7, Lines 165, even though the data has been shown in the figure, the reviewer suggested it may be easier to read if some key data was also shown up in the right place in the text.

(5) In Page 12, Lines 325, is it an incomplete writing?

Reviewer #3 (Remarks to the Author):

The authors have done an excellent job describing adsorbent media production for menstrual pads from Sisal. The experiments have been described in detail on mild delignification using performic acid, mechanical fluffing, physical and chemical properties of produced fibers, and absorption capacity of materials followed by carbon and water footprint analysis.

I suggest authors reread the manuscript for any spelling/grammatical errors.

Reviewer #1 (Remarks to the Author):

The manuscript submitted by Molina et al. shows a promising novel source of absorbent alternatives based on *Agave Sisalana* structures. The work has a high potential to become a reference in sustainable absorbent materials using novel bio-based sources. However, some improvements are needed to increase the article's impact and clarify the work's aim. Please, find here my point-by-point comments:

1) In the title, what do the authors mean by "distributed production"? The title should be clear, reflecting the main outcome of the work.

We thank the reviewer for bringing up this point and we have made efforts to be more specific about exactly what we mean by our reference to "distributed production."

Our use of the term "distributed production" is informed by our experience working with various NGOs and entrepreneurs who are operating small scale menstrual pad manufacturing facilities in areas that are currently underserved by existing supply chains. These small scale menstrual pad manufacturers provide an alternative to improvised solutions that are frequently unreliable and unhygienic. Here, production is "distributed" in the sense that it is operating at smaller scales and rooted more closely to the communities they are intended to serve. This means production is less vulnerable to supply chain disruptions, provides greater distribution of value-add production, and often provides access to more efficient distribution channels. Thus, to make this meaning clear and to highlight why we think this idea is important, we have added the following lines to our introduction which contain several references guiding the reader to more thorough descriptions of these ideas:

(Lines 38-41): In the context of this paper, distributed production is a manufacturing practice where production occurs in close geographic proximity to the communities being served [5]. Reduced economies of scale are often compensated by more resilient supply chains, simplified logistics, sustainability, and the possibility to share know-how between different production nodes globally using digital communication technologies [6, 7, 8].

Having provided a detailed description of our use of the term "distributed production", we now turn our attention to the reviewer's concern over the extent to which the title reflects the main outcome of the work.

The main outcome of this work shows that *Agave sisalana* can be readily processed into an absorbent material for use in absorbent hygiene applications. Since *Agave sisalana* is a crop that can grow in water-constrained climate, we have shown that material feedstocks for this application can be obtained in regions where raw materials would have to otherwise be sourced from global supply chains.

Moreover, the chemistry used to convert the *Agave sisalana* feedstock into a highly absorbent and retentive material is consistent with a distributed production framework. First, the two

consumable reagents (hydrogen peroxide and formic acid) can be produced in a scale-free manner via electrochemistry - as evidenced in literature and by emerging commercialization efforts. Second, the peroxyformic acid chemistry utilized in our work produces no effluent, Fenton decomposition ensures that the waste products consist of water and carbon dioxide¹. These two considerations significantly reduces the constraints associated with economies of scale in typical pulp production (typical paper mills operate at a scale of 400-1500 kton/year²) and concern over responsible management of effluent.

Finally, we make an extensive effort in the *Introduction* and *Discussion* sections to connect the proposed production process to the engineering challenge of providing new materials in support of menstrual pad manufacturing efforts in underserved markets. Our future work will test this process with field partners in Kenya and Nepal to understand the techno-economic viability of the distributed production approach described in this work. However, until such a demonstration is made, we concede that we have not shown distributed production but we believe that we have made an important step towards it. As such, we have amended our title:

(Title:) Agave Sisalana: towards distributed production of absorbent media for menstrual pads in semi-arid regions.

2) In the abstract, the 23 g/g refers to absorption or retention? These two are very distinct terms in superabsorbents and need to be clear. Also, only absorption under load (AUL) was assessed, while other important parameters, such as free swelling and centrifuge retention capacity, are not stated. These evaluations are critical for proposing new materials to replace current absorbent items such as single-use pads. Please check the EDANA standards (nonwovens) for more information.

This is an important distinction. Here we propose a high grade absorbent material but we are not categorizing the sisal-based absorbent as a “superabsorbent”. Superabsorbent materials used in menstrual pads are typically available in powder forms and do not provide a minimal foundation for constructing a pad.

For specific characterization tests, we chose a test based on the standard of the Scandinavian Pulp, Paper, and Board Testing Committee (SCAN-C-33-80)³. This standard has been used to evaluate absorption properties of fluff materials in the literature⁴. This standard measures the mass of absorbed liquid retained under load - assessing both absorption and retention. This test is well suited for the specific application for which we have been developing the materials (i.e. menstrual pads) presented in the current manuscript. From what we've found,

¹ Santacesaria, E., Russo, V., Tesser, R., Turco, R. & Di Serio, M. Kinetics of performic acid synthesis and decomposition. *Industrial Engineering Chemistry Research* 56, 12940–12952 (2017)

² Björjesson, M. H. & Ahlgren, E. O. Technology Brief I07: Pulp and Paper Industry (International Energy Agency - Energy Technology Systems Analysis Program, 2015)

³ Scandinavian Pulp, Paper, and Board Testing Committee. Fluff: specific volume and absorption properties. SCAN-C-33-80 (1980).

⁴ Azevedo, C. A., Rebola, S. M. C., Domingues, E. M., Figueiredo, F. M. L. & Evtuguin, D. V. Relationship between surface properties and fiber network parameters of eucalyptus kraft pulps and their absorption capacity. *Surfaces* 3, 265–281 (2020).

EDANA standards provide only broad guidelines for testing absorption performance in feminine hygiene products:

For sanitary pads and panty liners various methods are in use to measure the absorption capacity of the products. Various methods have been developed to assess the retention capacity (absorption before leakage). These methods range from simple designs (dunk, fluid acquisition) to those that take the shape and the features to prevent leakages into consideration⁵.

We acknowledge that our work with Sisal might find new applications in other industrial products and thus expanding the set of tests - suggested by the reviewer above - in the future is indeed very relevant.

3) Is it unclear to the reviewer how the work will contribute to increasing access to menstruation pads for women in medium/low-income countries? This needs to be clearer in the revised version.

Currently, 500 million women around the world lack access to sanitation and hygiene products⁶. Aside from high cost, many conventional pads are not available in remote areas either due to high cost or low commercial distribution by international companies due to, among other things, concern over disposal and lack of familiarity with distribution channels.

In response to this large unmet need, a large number of NGOs and social entrepreneurs are engaged in creating business to manufacture menstrual pads locally. As described in our manuscript, a key barrier is access to materials that can be sourced in-country - particularly sustainable materials. These entrepreneurs lack any access to research infrastructure and hence can not invent new materials based on locally available resources. Our work directly addresses this gap.

Since the current paper resolves one of the key missing materials required to produce a functioning menstrual pad, we emphasize the social and environmental impact associated with our proposed process. To explicitly address the reviewer's comment, we have included the sentence:

(Lines 356-357): This demonstration will increase access to materials used in menstrual pads by paving the way toward their production in semi-arid regions.

Additionally, we have edited the last line of our conclusions section to highlight the current work addresses one component of a multi-material problem and that additional work in this spirit is of great need to the community of NGOs and social entrepreneurs described above:

⁵ EDANA Guidelines for the Testing of Feminine Hygiene Products – version 13th December 2018

⁶ Amaya, L., Marcatili, J. & Bhavaraju, N. Advancing Gender Equity by Improving Menstrual Health: Opportunities in Menstrual Health and Hygiene (FSG, 2020)

(Lines 361-365): Additionally, complementary production strategies are still urgently needed to realize the production of a complete menstrual pad from sustainable material feedstocks. Our work presents a new manufacturing strategy for a key component of an essential product to be made in a distributed fashion, bringing further economic development in regions disadvantaged by climate.

Our ongoing work is addressing these additional materials with the intention of producing a fully functional menstrual pad using locally-available biomass.

4) Non-wood alternatives are stated in the introduction. However, the AUL obtained in this study needs to be compared with other non-wood options and/or synthetic superabsorbent polymers (SAP) used commercially today. The authors should put the work into a comparative perspective to increase the impact further.

The current manuscript already included comparisons to other non-wood options. We refer the reviewer to Figure 1k where we introduce cotton obtained from a commercially available menstrual pad as our reference. Additionally, in Figure 4l, we compare sisal to the common non-wood alternatives Flax and Hemp. There, we present absorption under pressure data for both raw fibers and fibers obtained after the mild delignification with peroxyformic acid described in our work.

Fig. 1(k)

Fig. 4(l)

Additionally, we have now included data on timber fluff pulp as well (15.69 g/g). The data are presented in Supplementary Fig. 8. See response to Comment #7.

We reiterate that our work is not proposing a new superabsorbent material and hence comparison to those materials would not be relevant. We are primarily focused on finding new absorption materials that allow local manufacturing of menstrual pads in semi-arid areas of the

world. This is why our work focuses on Sisal - a common drought resistant plant found across the world.

5) The authors mention non-wood alternatives in the introduction and discuss other lignocellulosic alternatives in the article text. However, what about other works pursuing alternative sources of absorbent polymers, e.g., proteins from agro-food biomass waste? A reflection on this could broaden the perspectives of the suggested materials. Some references are suggested here:

1. <https://onlinelibrary.wiley.com/doi/10.1002/jsfa.9738>
2. <https://doi.org/10.3390/polym15020351>
3. <https://doi.org/10.1038/s42004-021-00491-5>

We thank the reviewer for listing these papers. We do not focus on superabsorbent polymers because they are typically found as powders and used as an addition in absorbent hygiene products to reduce bulk while increasing retention. Thus, a superabsorbent powder alone does not provide a minimal foundation for constructing a menstrual pad.

However, based on this suggestion, we have included a description of some additional materials. We also describe the possibility of further increasing the performance of the non-wood fibrous materials that are the focus of the present manuscript by including references to the protein based superabsorbent polymers described by the reviewer:

(Lines 45-58): Recent advances in absorbent fiber materials have focused on materials produced from cellulose derivatives [14, 15]. However, these approaches require access to advanced timber products which might not be reliably available. Meanwhile, superabsorbent polymers (SAPs) are often included in absorbent hygiene products to increase absorption and retention while reducing the product's bulk [16, 17]. However, SAPs are produced from synthetic polymers with a high environmental footprint [18, 19]. Significant advances have been made in the production of biodegradable SAPs using proteins sourced from agricultural waste streams [20, 21]. However, to make a complete menstrual pad, these biodegradable SAPs must still be embedded in a fibrous matrix to provide structure and facilitate liquid transport [17, 22]. Taken together with the other materials required for producing a disposable menstrual pad (e.g. porous top layer and waterproof back layer), conventional products represent a significant sustainability challenge in terms of plastic waste [23, 19], health effects [24], and their burden on sanitation systems [1, 25]. Thus a key challenge in realizing the distributed production of menstrual pads will be to develop the necessary functional functional materials for their construction utilizing locally-sourced regenerative materials [26].

6) How sustainable is the delignification process suggested? Are the reagents recirculated? How is the process thought on a large scale, and what are the possible side-streams generated? How cost-competitive are these materials compared to non-wood alternatives and synthetic SAP materials?

We have evaluated the sustainability of the delignification process in several ways. The current manuscript includes a cradle-to-gate life cycle analysis assessing the carbon footprint and water consumption of the proposed process. We find that our delignification process has a similar carbon footprint to values reported for bleached fluff pulp production (see Figure 5g and 5h) and a slightly higher water consumption rate. Compared with cotton, the carbon footprint associated with our proposed process is significantly improved when harvest and collection activities are taken into account.

A unique feature of our work is that we assess the carbon footprint attributed to transportation of materials within global supply chains. We have defined the "post-gate" scope in Figure 5e where we show that the emissions associated with transportation are significant. We have gathered from literature several transportation different scenarios (see Figure 5g inset, Supplementary Note 6, and Supplementary Tables 6-8). This scope is not often included in life cycle analyses but is often included as an important advantage associated with distributed and/or localized production. An important aspect of this work is to explicitly quantify the carbon footprint associated with a distributed production approach.

Regarding recirculation of reagents, we have already made several remarks on this point in our original submission. For example, in the *Introduction*:

(Lines 87-97): In the case of peroxyformic acid, decomposition occurs rapidly into water and carbon dioxide, eliminating the introduction of adsorbable organic halides into the environment [50]. Meanwhile synthetic systems that allow for recycling of reagents have also emerged based on solid di-carboxylic acids [51], deep eutectic solvents [52], and organosolv pulping [53, 36]. Thus, there are two conceptual approaches to operating the pulping process that minimize dependence on an external chemical supply chain: recycling and on-site production. If the reagents can be recycled, energy must be expended to recover them. If reagents are consumed, then they must be efficiently produced on-site. Our study is motivated by the increasing capacity for on-site production of chemicals like hydrogen peroxide [54, 46] and formic acid [55, 56, 57]. Few studies have investigated the use of these technologies to implement a bioinspired strategy to transform lignocellulosic biomass into absorbent media in an environmentally sustainable way at small scales.

And again in the the section titled *Discussion and conclusions*:

(Lines 336-338): More broadly, this is a chemical strategy that exists amongst many other possible alternatives such as recyclable chemicals, solid di-carboxylic acids, and enzymes.

However, understanding these numerous trade offs must be considered within a comprehensive techno-economic framework.

Further, we maintain that one of the key findings of this work is that fibers derived from *Agave sisalana* itself is a highly performant material. We suspect and state in our submission that other chemistries might also be viable. However, their sustainability must be evaluated within context-specific techno-economic analysis. Our life cycle analysis does not assume any recirculation.

Regarding generation of possible side streams, we reiterate that peroxyformic acid readily decomposes into water and carbon dioxide. There is a waste stream associated with decortication of sisal which is accounted for in our life cycle inventory by way of the excellent research by Broeren *et al.*⁷ Additionally, there is a waste stream associated with the removed lignin. Characterization of this waste product is beyond the scope of the present research. However, given the growing use of peracid chemistry to remediate waste water at scale, we maintain that the process does not generate any appreciable effluent. Further, we have already referenced both of these possibilities in the original submission:

(Lines 331-336): Developing strategies to valorize these waste streams will be important since they might enable the production of additional products. For example, waste lignin might be useful for producing compostable barrier sheets [52] while other components might be useful as fertilizer [84]. The former will enable the production of complete products such as menstrual pads using regionalized biomass while the latter enables a regenerative bioeconomy [26].

Regarding cost competitiveness, decorticated Sisal fiber is available in Kenya at \$800/ton. A superficial cost analysis would be of little use. A detailed cost analysis is challenging since it is difficult to include hidden costs associated with a context specific implementation and will be a goal of the next stage of our work which will focus much more explicitly on scaling. But we are confident that the abundant availability of Sisal makes this a competitive material feedstock for menstrual pad manufacturing. Direct cost comparisons to SAPs (synthetic or natural) is not relevant for this current manuscript. However, we will explore the role of natural SAPs to further improve the retention properties of our final products.

⁷ Broeren, M. L. M. & et al. Life cycle assessment of sisal fiber – exploring how local practices can influence environmental performance. *Journal of Cleaner Production* 149, 818–827 (2017).

7) Line 272, again, the results are compared with bleached cellulose fluff pulp, but what about the absorption values of this material and also compared to synthetic counterparts?

We have now included the absorption value of bleached fluff pulp obtained from two different sources as shown in Supplementary Figure 8:

Supplementary Figure 8: Baseline absorption Bar chart comparing the absorption capacities of cotton CMP with timber fluff pulp obtained from two sources that are used in commercially-available-pads

We reiterate: since we do not focus on superabsorbent materials in this paper, we have not included any of them in the comparison. Instead we have based our work on comparison with cotton and other commonly used wood-derived products.

8) Line 325, what is that sentence?

That was an overlooked sentence. Line 325 has been removed.

9) Line 383, which liquid are the authors referring to? The results can vary greatly between different liquids of interest for the sanitary area. For instance, pure water vs. saline solution (usual industrial standard) can produce very different absorptions due to osmotic pressure effects. More information can be found at: <https://doi.org/10.1021/acssuschemeng.8b05400>

Thank you for bringing this to our attention. We are referring to the test liquid described in Indian Standard 5405:1980. To clarify this point we have included a description of its composition in the section titled *Absorption under pressure testing*:

(Lines 404-407): Absorption is assessed using a test solution prepared according to Indian Standard 5405:1980 [66]. Briefly, 400 mg of methyl paraben and 74 g of gum arabica are added

to 600 mL of boiling water. Once cooled, 900 mg of methylene blue and 147 mL of glycerin are added and the final volume is adjusted with water to 920 mL. The solution was mixed thoroughly and allowed to stand at least 24 hours. Prior to the absorption test, each test square was weighed (Wi) and its average height measured using a caliper.

10) Line 384, is 5 min a representative value for absorption in sanitary applications? Normal values are around 30 min.

The work presented in the paper involved screening and testing a large number of materials. Thus, to perform these combinatorial experiments, extended soaking times per test would not have been viable. In our initial experiments, we experimentally assessed the difference between an extended soaking period (30 min.) and the time used in this work (5 min.) and the difference in results was small: 15.00 g/g versus 15.92 g/g or about 6% which is within the error of our measurements. This difference was already highlighted in our original manuscript:

(Lines 412-413): The test square is left to absorb for 5 minutes; longer absorption times were evaluated but no significant differences were measured after 5 min.

11) What is the intention behind assessing the static contact angle for an absorbing material?

This is a good question. Absorption in a fiber network depends on two factors: fiber-liquid interactions and the geometric structure of the fiber network. The fiber-liquid interaction can be characterized by measuring the static contact angle⁸. By directly measuring this angle in microscopic fibers - we are able to understand to what extent absorption depends on these two factors. This is a common test for fibrous materials - see Hodgson and Berg for a nice discussion of these measurements⁹. These considerations have been described in our paper in the section titled *Design criteria and evaluation* but we have added a reference to the same to provide additional clarity:

(Lines 125-126) Specifically, absorption in a fiber network depends on two factors: fiber-liquid interactions and the geometric structure of the fiber network [64].

⁸ Young, R. Wettability of wood pulp fibers: Applicability of methodology. *Wood Fiber Science* 8, 120–128 (1976)

⁹ Hodgson, K. T. and Berg, J. C. Dynamic Wettability Properties of Single Wood Pulp Fibers and Their Relationship to Absorbency. *Wood Fiber Science* 20, 1-17 (1988)

12) Figure 2, why does the color of some materials after absorption look blue, black, and mixture?

The change in color is associated with the amount of test liquid absorbed. Samples that have imbibed a large amount of fluid appear nearly black whereas those that have imbibed significantly less appear blue. For clarity, we have mentioned this observation in the section titled *Delignification of sisal fibers*:

(Lines 224-225): However, there is an appreciable increase following alkali treatment, which is evidenced by the increasing intensity of absorbed test liquid following immersion such that the alkali-washed sample appears nearly black (Figure 2e-g)

13) Figure 2, FTIR is missing the X-axis units.

We have added the appropriate units (wavenumber [cm^{-1}]) to the x-axes of Figure 2h:

14) The scale of the microCT in Figure 3 needs to be included. Please add the scale bar.

We have added the appropriate scale bar in Figure 3n-p:

Reviewer #2 (Remarks to the Author):

This manuscript evaluated the absorbing performance of delignified sisal fibers in the application of menstrual pads. The reviewer believes this study is very essential for the woman hygiene improvement and sustainability development in local communities. Minor revision is recommended prior to publication. Please see the comment details below.

(1) In Page 6 Line 144, it may be easier to understand when a detailed data is added here.

Thanks for this suggestion. We have now included the detailed data in the text:

(Lines 161-163): The resulting fibers show only a slight increase in absorption capacity (7.53 g/g) compared with the untreated reference (5.35 g/g) (Figure 1k, Supplementary Note 3). Further, the fibers retain an uncomfortable coarse texture which make them unsuitable for application in a sanitary pad.

(2) In Page 7 Lines 159, please add citation here.

We have included two citations. First, a study detailing the potential radical species generated by a closely related peracid system (peracetic acid) and their relevance for degrading organic wastewater contaminants¹⁰. Second, an excellent review by More *et al.* describing oxidation reactions of lignin by hydrogen peroxide detailing the diversity of potential reactive oxygen species and possible reactions by way of model compound studies¹¹.

(Lines 177-179) The exact details of the reaction pathways are difficult to know and are out of the scope of the present research, since they depend on both the identity of the reactive oxygen species and the composition of the lignins involved [48, 74].

(3) In Page 7, Lines 163, maybe it will be better to add yield data here to help describe.

Thanks for this comment. We have included the yield data from the 1% peroxyformic acid treatment which supports the claim that lower concentrations of peroxyformic acid leave residual lignin on the fibers:

(Lines 181-183): Lower concentrations (1%) of peroxyformic acid gave microfibers that retain some brown color at a yield of 77% (w/w) (Supplementary Fig. 6). The color and higher yields suggest appreciable amounts of residual lignin are retained by the material.

(4) In Page 7, Lines 165, even though the data has been shown in the figure, the reviewer suggested it may be easier to read if some key data was also shown up in the right place in the text.

¹⁰ Cai, M., Sun, P., Zhang, L. & Huang, C.-H. Uv/peracetic acid for degradation of pharmaceuticals and reactive species evaluation. *Environmental Science Technology* 51, 14217–14224 (2017).

¹¹ More, A., Elder, T. & Jiang, Z. A review of lignin hydrogen peroxide oxidation chemistry with emphasis on aromatic aldehydes and acids. *Holzforchung* 75, 806–823 (2021)

Thanks again for this suggestion. We have now included the numbers in the text as well:

(Lines 179-186): High concentrations (10%) of peroxyformic acid yield sisal microfibers that appear completely bleached with a yield of 60% (w/w) (Supplementary Fig. 5) and high absorption performance (23.07 g/g). Lower concentrations (1%) of peroxyformic acid yield microfibers that retain some brown color at a yield of 77% (w/w) (Supplementary Fig. 6). The color and higher yields suggest appreciable amounts of residual lignin are retained by the material. Despite the presence of some residual lignin, the absorption performance (15.52 g/g), while reduced compared with the totally delignified samples, is still competitive with cotton-CMP (15.19 g/g) (Fig. 1k) and bleached wood-derived fluff pulps (15.69 g/g) (Supplementary Fig. 8).

(5) In Page 12, Lines 325, is it an incomplete writing?

That was an overlooked sentence. Line 325 has been removed.

Reviewer #3 (Remarks to the Author):

The authors have done an excellent job describing adsorbent media production for menstrual pads from Sisal. The experiments have been described in detail on mild delignification using performic acid, mechanical fluffing, physical and chemical properties of produced fibers, and absorption capacity of materials followed by carbon and water footprint analysis.

I suggest authors reread the manuscript for any spelling/grammatical errors.

We thank the reviewer for supportive comments and have reviewed the manuscript for any further spelling and/or grammatical errors.

REVIEWERS' COMMENTS:

Reviewer #3 (Remarks to the Author):

The authors have significantly improved the manuscript. Hence, it is ready for publication.

Reviewer #4 (Remarks to the Author):

Nice work! The Authors have addressed all problems.